# Manuka-type honeys can eradicate biofilms produced by *Staphylococcus aureus* strains with different biofilm-forming abilities

Jing Lu[1], Lynne Turnbull[1], Catherine M. Burke[1], Michael Liu[1], Dee A. Carter[2], Ralf C. Schlothauer[3], Cynthia B. Whitchurch[1] and Elizabeth J. Harry[1]

[1] The ithree institute, University of Technology Sydney, Sydney, NSW, Australia
[2] School of Molecular Bioscience, University of Sydney, Sydney, NSW, Australia
[3] Comvita NZ Limited, Te Puke, New Zealand

## ABSTRACT

Chronic wounds are a major global health problem. Their management is difficult and costly, and the development of antibiotic resistance by both planktonic and biofilm-associated bacteria necessitates the use of alternative wound treatments. Honey is now being revisited as an alternative treatment due to its broad-spectrum antibacterial activity and the inability of bacteria to develop resistance to it. Many previous antibacterial studies have used honeys that are not well characterized, even in terms of quantifying the levels of the major antibacterial components present, making it difficult to build an evidence base for the efficacy of honey as an antibiofilm agent in chronic wound treatment. Here we show that a range of well-characterized New Zealand manuka-type honeys, in which two principle antibacterial components, methylglyoxal and hydrogen peroxide, were quantified, can eradicate biofilms of a range of *Staphylococcus aureus* strains that differ widely in their biofilm-forming abilities. Using crystal violet and viability assays, along with confocal laser scanning imaging, we demonstrate that in all *S. aureus* strains, including methicillin-resistant strains, the manuka-type honeys showed significantly higher anti-biofilm activity than clover honey and an isotonic sugar solution. We observed higher anti-biofilm activity as the proportion of manuka-derived honey, and thus methylglyoxal, in a honey blend increased. However, methylglyoxal on its own, or with sugar, was not able to effectively eradicate *S. aureus* biofilms. We also demonstrate that honey was able to penetrate through the biofilm matrix and kill the embedded cells in some cases. As has been reported for antibiotics, sub-inhibitory concentrations of honey improved biofilm formation by some *S. aureus* strains, however, biofilm cell suspensions recovered after honey treatment did not develop resistance towards manuka-type honeys. New Zealand manuka-type honeys, at the concentrations they can be applied in wound dressings are highly active in both preventing *S. aureus* biofilm formation and in their eradication, and do not result in bacteria becoming resistant. Methylglyoxal requires other components in manuka-type honeys for this anti-biofilm activity. Our findings support the use of well-defined manuka-type honeys as a topical anti-biofilm treatment for the effective management of wound healing.

Corresponding author
Elizabeth J. Harry,
liz.harry@uts.edu.au

## INTRODUCTION

Chronic wounds currently affect 6.5 million people in the US. These wounds are difficult to treat and estimated to cost in excess of US $25 billion annually, with significant increases expected in the future (*Sen et al., 2009*). A wound is generally considered chronic if it has not started to heal by four weeks or has not completely healed within eight weeks (*McCarty et al., 2012*). Such prolonged, non-healing wounds are caused by a variety of factors, with bacterial infection being a significant contributor.

In chronic wounds, as with everywhere on earth, bacterial cells predominantly exist as biofilms, where cells are embedded within a matrix of polysaccharides and other components. This matrix affords resistance to environmental stresses such as altered pH, osmolarity, and nutrient limitation (*Fux et al., 2005*). The matrix also limits access of antibiotics to the biofilm embedded cells (*Ranall et al., 2012*), which are up to 1,000 times more recalcitrant to these compounds than planktonic cells (*Hoyle & Costerton, 1991*). Planktonic bacteria may also contribute to pathogenesis, as their release from biofilms has been proposed to maintain the inflammatory response within the wound (*Ngo, Vickery & Deva, 2012*; *Wolcott, Rhoads & Dowd, 2008*), as well as allowing seeding to other areas (*Battin et al., 2007*; *Costerton et al., 2003*). Along with the difficulties of treating biofilm infections, the emergence of resistance to multiple antibiotics has exacerbated the problem of chronic wound treatment (*Engemann et al., 2003*; *Projan & Youngman, 2002*). Thus, there is an increasing need for new approaches to combat bacterial biofilms in chronic wounds.

Honey has been used to treat acute and chronic wound infections since 2500 BC (*Forrest, 1982*; *Molan, 1999*; *Simon et al., 2009*). Honey possesses a number of antimicrobial properties including high sugar content, low pH, and the generation of hydrogen peroxide by the bee-derived enzyme glucose oxidase (*Stephens et al., 2009*). However, not all honeys are the same and their antimicrobial properties vary with floral source, geographic location, weather conditions, storage (time and temperature) and various treatments, such as heat (*Al-Waili et al., 2013*; *Allen, Molan & Reid, 1991*; *Molan, 1999*; *Sherlock et al., 2010*). These factors lead to differences in the levels of various antibacterial components. Manuka honey is derived from *Leptospermum scoparium* bush and is particularly potent (*Adams et al., 2008*; *Allen, Molan & Reid, 1991*; *Kwakman et al., 2011*). This is believed to be largely due to the high levels of the reactive dicarbonyl methylglyoxal (MGO) (*Adams et al., 2008*; *Mavric et al., 2008*), which is highly inhibitory to bacterial growth (*Lu et al., 2013*). Other antimicrobial compounds in honeys include bee defensin-1 (*Kwakman et al., 2010*; *Kwakman & Zaat, 2012*), various phenolic compounds and complex carbohydrates (*Adams et al., 2008*; *Gresley et al., 2012*; *Mavric et al., 2008*; *Molan, 1999*; *Weston, Brocklebank & Lu, 2000*). The combination of these diverse assaults may account for the inability of bacteria to develop resistance to honey (*Blair et al., 2009*; *Cooper et al., 2010*), in contrast to the rapid induction of resistance observed with conventional single-component antibiotics (*Colsky, Kirsner & Kerdel, 1998*; *Cooper, 2008*).

A few studies have examined the effect of manuka honey on biofilms, showing it to be active against a range of bacteria, including Group A *Streptococcus pyogenes*, S*treptococcus mutans, Proteus mirabilis, Pseudomonas aeruginosa, Enterobacter cloacae* and *Staphylococcus aureus* (*Alandejani et al., 2008*; *Maddocks et al., 2013*; *Maddocks et al., 2012*; *Majtan et al., 2013*). However, the levels of reported anti-biofilm activity are not consistent among these studies. This is highly likely to be at least in part due to differences in the levels of the principle antibacterial components in the honey, MGO and hydrogen peroxide, which varies with the floral and geographic source of nectar, the honey storage time and conditions, and any possible other treatments that may have occurred. All these conditions affect the antimicrobial activity of honey (*Adams et al., 2008*; *Al-Waili, Salom & Al-Ghamdi, 2011*; *Sherlock et al., 2010*; *Stephens et al., 2009*), but are often not reported. Importantly, medical-grade honeys, while often composed primarily of manuka, can also contain honey derived from other flora sources, which can alter the levels of various antimicrobial components. Therefore, it is imperative to use well-characterized honeys to enable both accurate comparisons among studies, and the rigorous assessment of the potential of medical-grade honey to be used in wound treatment in the clinic.

Here we have performed biomass and viability assays, as well as confocal scanning light microscopy to examine the anti-biofilm activity of four NZ manuka-type honeys, clover honey and an isotonic sugar solution on a range of *S. aureus* strains that differ widely in their biofilm-forming ability. These honeys have been well characterized in terms of their geography, floral source and the level of the two principal antibacterial components found in honey, MGO and hydrogen peroxide. We demonstrate that the manuka-type honeys are highly active in both the prevention and elimination of methicillin-sensitive and methicillin-resistant *S. aureus* biofilms. The antibiofilm activity was highest in the honey blend that contained the highest level of manuka-derived honey; although the same level of MGO, with or without sugar, could not eradicate biofilms. This suggests that additional factors in these manuka-type honeys are responsible for their potent anti-biofilm activity; and emphasise the importance of characterizing honey in order to understand and choose the best honey product for wound management.

## MATERIALS AND METHODS

### Honey samples

The New Zealand (NZ) honey samples used in this study are listed in Table 1, and include monofloral manuka honey, Medihoney (a manuka-based medical grade honey; Comvita NZ Ltd), a manuka/kanuka blend, and clover honey (a white New Zealand honey). All honey samples were supplied by Comvita NZ Ltd. (Te Puke, New Zealand). The harvesting and geographic information for these honeys, as well as the levels of the three major antimicrobial components: methylglyoxal (MGO), di-hydroxyacetone (DHA) and hydrogen peroxide, are listed in Table 1. All samples were stored in the dark at 4 °C and were freshly diluted in Tryptone Soya Broth (TSB) immediately before use in assays. All honey concentrations are expressed as % w/v.

**Table 1  Harvesting and chemical information for the tested NZ honey samples.**

| Honey type | Harvest period | Area | Floral source | Major antimicrobial composition | | |
| --- | --- | --- | --- | --- | --- | --- |
| | | | | DHA[a] | MGO[a] | $H_2O_2$[b] |
| **Manuka** | Spring 2010 | Hokianga, Northland, NZ | *Leptospermum scoparium var. incanum* | 4277 | 958 | 0.34 |
| **Medihoney** | Spring 2010 | Northland, NZ | *Leptospermum scoparium var. incanum + Kunzea ericoides* | 883 | 776 | 0.31 |
| **Manuka/kanuka** | Summer 2010/11 | Hokianga, Northland, NZ | *Leptospermum scoparium var. incanum + Kunzea ericoides* | 652 | 161 | 0.68 |
| **Clover** | N/A[*] | Balcutha, Otago, NZ | *Trifolium spp.* | <20 | <10 | 0.11 |

Notes.

[a] MGO (methylglyoxal) levels were analyzed against di-hydroxyacetone (DHA) and expressed as mg MGO per kg of honey.

[b] Rate of production of $H_2O_2$ (hydrogen peroxide) is expressed as μmol/h in 1 mL of 10% w/v honey.

[*] Information not available.

## Other tested solutions

A series of other solutions were included for investigation alongside the honey samples: (i) a sugar solution designed to mimic the concentration and composition of honey sugars (45% glucose, 48% fructose, 1% sucrose) diluted as above for honey; (ii) MGO diluted in TSB to concentrations similar to those present in the manuka-type honeys (100 mg/kg, 700 mg/kg and 900 mg/kg honey) to assess the effect of MGO alone on bacterial growth; (iii) MGO diluted in sugar solution to the same concentration as (ii). MGO was obtained as a ∼40% (w/w) solution in water (Sigma-Aldrich Co., MO, USA).

## Hydrogen peroxide assay

The level of hydrogen peroxide produced by the NZ honey samples was determined using a hydrogen peroxide/peroxidase assay kit (Amplex Red; Molecular Probes, Life Technologies Corp., Carlsbad, CA, USA) as previously reported (*Lu et al., 2013*).

## Bacterial strains and growth conditions

Four strains of *S. aureus* were examined. These include two laboratory reference strains: NCTC 8325 (National Collection of Type Cultures) (*Stepanovic et al., 2000*) and ATCC 25923 (American Type Culture Collection) which are methicillin-sensitive, and two clinical isolates: MW2 (Hospital-Acquired Methicillin-resistant *Staphylococcus aureus*, HA-MRSA) (*Baba et al., 2002*) and USA300 (Community-Acquired Methicillin-resistant *Staphylococcus aureus*, CA-MRSA) (*Kazakova et al., 2005*). All *S. aureus* strains were grown in TSB at 37 °C. For optimal biofilm formation, 1% (w/v) glucose was added to this medium (TSBG) except for strain NCTC 8325 which was found to produce optimal biofilm in the absence of added glucose.

## Susceptibility of *S. aureus* to NZ honeys: growth response assays

In this study, growth response assays were carried out to assess whether the NZ honeys affected cell growth of the different strains of *S. aureus* (at concentrations of 1–32%;

prepared as serial 2-fold dilutions in TSB(G)). Details of the growth assay methods are described in our previous publication (*Lu et al., 2013*). TSB(G) media without honey was included as a control. Unless otherwise stated, all assays were performed with three biological replicates and three technical repeats of each replicate.

## Biofilm formation assays

The effects of NZ honeys and other solutions on *S. aureus* biofilm formation were determined using crystal violet static biofilm formation assays in microtitre plates according to published studies with the following modifications (*Christensen et al., 1985*; *Stepanovic et al., 2000*). Crystal violet stains all biomass including live and dead cells and the biofilm matrix. *S. aureus* strains were cultured in 2 mL of TSB(G) with shaking (250 rpm) overnight at 37 °C . A suspension from the overnight culture was then diluted to a cell density of approximately $10^7$ CFU/mL in fresh TSB containing the appropriate test solution (honey, sugar solution, MGO or MGO in combination with sugar) to give a final volume of 150 μL. The suspension was added to each well of a 96-well tissue-culture treated microtitre plate (BD Falcon, NJ, USA). Media-only and media with the appropriate test solution without *S. aureus* inoculation were included as negative controls. The microtitre plates were sealed with AeraSeal (Excel Scientific, CA, USA) and incubated in a humidified incubator for 24 h at 37 °C. Following this, planktonic cell growth was assessed by transferring the planktonic phase into a new 96-well microtitre plate and reading the optical density at 595 nm with a microplate reader (VersaMax, Molecular Devices, California, USA). This step was required as *S. aureus* forms biofilms on the bottom of the microtitre plate wells, which interferes with optical density readings of the planktonic culture. The microtitre plates with residual biofilm were then washed three times with sterile phosphate buffered saline (PBS) to remove unattached cells and air-dried at 65 °C for 1 h, to fix the *S. aureus* biofilm to the bottom of the well surface. The plate was then stained with 0.2% (w/v) crystal violet at room temperature for 1 h, excess crystal violet solution was decanted and the plates were washed as above with PBS. Stain that was bound to the adherent biomass was resolubilized with 200 μL 33% acetic acid and transferred into a new 96-well microtitre plate to measure the $OD_{595}$.

## Biofilm elimination assays

*S. aureus* biofilms were first formed in the wells of a 96-well microtitre plate for 24 h at 37 °C as described above. Biofilms were then washed three times with PBS. Various concentrations (0%–32% in two-fold serial dilutions) of honey and other test solutions were then added to the established *S. aureus* biofilms. The assay plates were then incubated for a further 24 h at 37 °C, and planktonic cell growth and biofilm mass were quantified as described above.

## Determination of bacterial cell viability in biofilms

Crystal violet stains all the components of the biofilm (*Bauer et al., 2013*). To quantify the viability of cells within the *S. aureus* biofilms following honey treatment, we used a BacTitre Glo Microbial Cell Viability Assay Kit (Promega, WI, USA), which measures

ATP levels as a proxy for viability. The assay reagents lyse the bacterial cells to release intracellular ATP, the levels of which are quantified via a luminescence-based luciferase activity assay (*Haddix et al., 2008*; *Junker & Clardy, 2007*). The BacTitre Glo protocol involved the same steps as crystal violet staining (above), however, instead of drying and staining the biofilms, plates were incubated with BacTitre Glo reagent in TSB(G) for 10 min at 37 °C in the dark. The contents of each well were then transferred into white solid-bottom 96-well microtitre plates (Cellstar; Greiner Bio-one, France) for luminescence measurement. Luminescence, which is proportional to the amount of ATP produced by metabolically active cells, was recorded using a 96-well microplate reader (Infinite 200Pro; TeCan, Männedorf, Switzerland).

To ensure the validity of this assay, a standard curve was constructed to assess the correlation between bacterial cell numbers and the luminescent signal in the biofilm. This was performed on the untreated control (containing *S. aureus* in TSB(G) only). Biofilms produced as above were washed and cells within the biofilm dispersed using a small probe sonicator (Sonics and Materials VC-505) to enable quantification by direct enumeration (*Merritt, Kadouri & O'Toole, 2005*). The recovered cell suspension was serially diluted 10-fold and a 20 μL aliquot was plated on Tryptone Soya Agar (TSA) for CFU determination. Luminescence of cells in the remaining suspension was assessed using the BacTitre Glo kit. From this, a correlated standard curve was constructed between calculated CFU/well and the relative luminescence readings. According to the standard curve shown in Fig. 1, the detection limit of the BacTitre Glo is at a luminescence reading below 1,000, which is equivalent to $10^3$ CFU/well (linear range from $10^3$–$10^7$ CFU/well). An upper limit was not detected.

## Visualizing live/dead stained *S. aureus* biofilms using confocal laser scanning microscope (CLSM)

*S. aureus* biofilms were treated with TSB containing 1%, 2%, 16%, and 32% NZ honeys or sugar solution for 24 h in black polystyrene 96-well microtitre plates with μClear bottoms (Cellstar; Greiner Bio-One, France) as described above, except the biofilm mass was not fixed by air-drying. The treated biofilm mass was washed three times with PBS and cells within the biofilm structure were fluorescently stained with 2.5 μM Syto9 (Invitrogen, CA, USA) and 4.3 μM propidium iodide (PI) (Becton Dickinson, NJ, USA), which identify live and dead cells in the biofilm structure, respectively. After 30 min of incubation in the dark at room temperature, the wells were washed thoroughly with PBS and fixed with 4% paraformaldehyde (Sigma-Aldrich, MO, USA) for 15 min. The wells were then rinsed and stored in PBS for imaging. Biofilms were imaged using confocal laser scanning microscopy imaging (CLSM) on a Nikon A1 confocal microscope. The Syto9 and PI fluorophores were excited at 488 nm and 561 nm, and the emissions were collected at 500–550 nm and 570–620 nm, respectively. For quantitative analysis, at least eight separate CLSM image stacks of each NZ honey treated biofilms were acquired with a resolution of 512 × 512 pixels. Biofilm biomass was calculated using COMSTAT (*Heydorn et al., 2000*) and is expressed as volume of the biofilm over the surface area ($\mu m^3/\mu m^2$). Representative

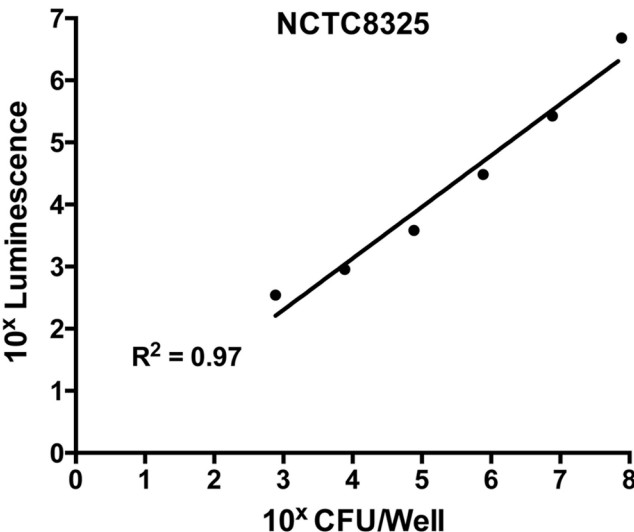

**Figure 1** **Correlation of levels of intracellular ATP to colony forming units (CFU) in static biofilms of** *S. aureus.* Static biofilms of *S. aureus* were formed in the wells of a microtitre plate for 48 h (with media replenishment at 24 h). After removal of the biofilm from the wall of each well, intracellular ATP levels were measured by the BacTitre Glo Viability Kit and CFU were determined for each well. The intracellular levels of ATP are plotted as a function of CFU and validate that the BacTitre Glo Viability Kit can be used as a surrogate measure of biofilm cell viability in subsequent assays.

presentation image stacks of each treatment were acquired at a resolution of 1024 × 1024 pixels and three dimensional biofilm images reconstructed using NIS-elements (Version10, Nikon Instruments Inc., USA). It should be noted that due to the incomplete displacement of Syto9 by propidium iodide in dead cells that there will remain some Syto9 staining of dead cells. Therefore the absolute level of live cells detected in the Syto9 channel will be somewhat overestimated using this co-staining method (*Stocks, 2004*).

## Assaying honey resistance in cells recovered from biofilms

The development of resistance is a great concern in clinical settings, where bacteria can become resistant to inhibitory compounds after exposure to sub-inhibitory concentrations (*Cars & Odenholt-Tornqvist, 1993*; *Pankuch, Jacobs & Appelbaum, 1998*). Planktonic cells that appeared after 24 h manuka-type honey treatment of established biofilms were assumed to have been released from the biofilm matrix. Cells recovered from biofilms treated with sub-eliminatory concentrations of manuka-type honeys were collected and tested for their ability to grow and form biofilms under the static growth conditions described above. Cell growth and biofilm formation were defined as not detected when the $OD_{(x)}-OD_{(media only blank)} \leq 0.1$. Each experiment was performed with three biological replicates and three technical repeats of each biological replicate.

## Statistical analysis

Statistical analysis to determine significant differences between treatments and among honey samples were performed. First, data sets were checked for normality (Gaussian Distribution) using the D'Agostino-Pearson normality test (alpha = 0.05) in GraphPad

Lu et al. (2014), *PeerJ*, DOI 10.7717/peerj.326

**Table 2  Concentration of honey required to inhibit *S. aureus* growth.**

| Honeys | NCTC 8325 | ATCC 25923 | MW2 | USA300 |
|---|---|---|---|---|
| Manuka honey | 8[*] | 8 | 8 | 8 |
| Medihoney | 8 | 8 | 8 | 8 |
| Manuka/kanuka honey | 16 | 16 | 16 | 16 |
| Clover honey | 32 | 32 | 32 | 32 |
| Sugar solution | >32 | >32 | 32 | >32 |

**Notes.**

[*] All numbers in the table are honey concentrations (%, w/v).

Prism (versions 5 and 6). Once normality was confirmed, significant differences between treatments and among honey samples were performed using One-Way ANOVA with Tukey Test using the same software. Statistical significance was set at $p < 0.05$.

# RESULTS

## The effect of NZ manuka-type honeys on the planktonic growth of *S. aureus*

Planktonic growth and biofilm mass were assessed to examine the ability of four NZ honeys, three manuka-types and one clover, and a sugar solution to prevent biofilm formation by different strains of *S. aureus*. Following 24 h incubation under static conditions, *S. aureus* cells formed biofilms at the bottom of microtitre plates, with very little or no planktonic growth detected, indicating that the concentration of honey needed to prevent *S. aureus* planktonic cell growth could not be calculated under these conditions. Shaking broth cultures were instead used to assess the effect of the treatments on planktonic growth. The results are shown in Table 2. Planktonic growth of the four *S. aureus* strains, NCTC 8325, ATCC 25923, MW2 and USA300, was completely inhibited by 8% manuka honey and Medihoney, 16% manuka/kanuka honey and 32% clover honey. The 32%, sugar solution was only effective at inhibiting growth of the MRSA strain MW2, with no inhibition of growth of the other strains at the concentrations tested (1–32%). These data are in agreement with the results of our previous study using the standard *S. aureus* reference strain, ATCC 25923, which used a similar suite of honey types and the same experimental conditions (*Lu et al., 2013*).

## The effect of NZ manuka-type honeys on *S. aureus* biofilm formation

All strains of *S. aureus* were assessed for their biofilm-forming ability after 24 h and 48 h. Under static conditions, biofilm-forming ability varied between strains, with NCTC 8325 forming the most robust biofilms, and generating significantly more biofilm mass than the other three tested strains (Figs. 2A and 2B). This was followed by ATCC 25923 and USA300, with MW2 forming the thinnest biofilms (Figs. 2A and 2B; $p < 0.05$). The effects of the four NZ honeys and the sugar solution on biofilm formation of strain NCTC 8325 are shown in Fig. 3A. Manuka honey was the most effective at preventing biofilm formation by *S. aureus* NCTC 8325, resulting in ~95% reduction ($p < 0.001$) in biofilm

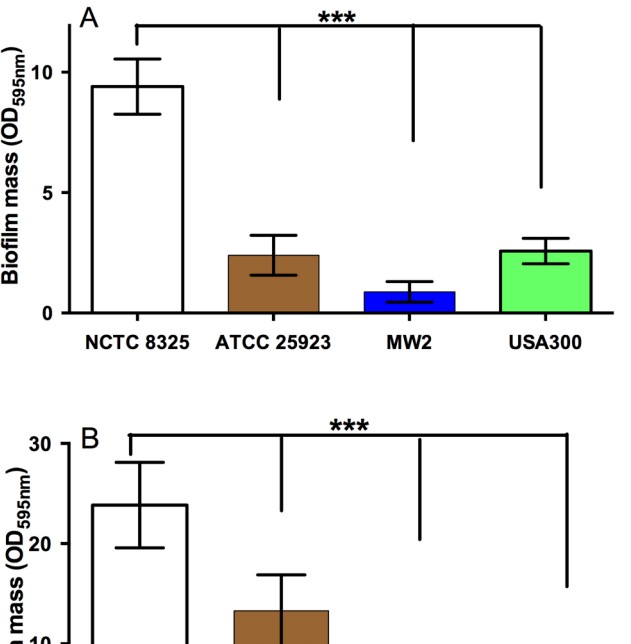

**Figure 2 Quantification of biofilm formation by different strains of *S. aureus.*** The ability of different strains of *S. aureus* to form biofilms on the plastic surface of tissue-culture treated 96-well microtitre plate was assessed in TSB(G) at 24 h and 48 h. Biofilm adherence was determined using a static biofilm formation assay over 24 h (A) and 48 h (with media replenished after 24 h incubation) (B). Biofilm formation was quantified by staining with 0.2% crystal violet solution and measured at an optical density of 595 nm. Error bars represent ±standard deviation (SD) of three biological samples performed in triplicate, *** represents $p < 0.001$, compared to NCTC 8325, as assessed by One-Way ANOVA with Tukey test after confirming normality of the data set for each treatment using the D'Agnostino-Pearson normality test.

formation at 8% (Fig. 3A) compared to the untreated (0%) control. At this concentration, the other honeys and the sugar solution did not significantly reduce biofilm formation. Medihoney and manuka/kanuka honey were highly effective at 16%, preventing biofilm formation by ~95% ($p < 0.001$). Clover honey was much less active and was less able to prevent biofilm formation than the sugar solution, even at the highest concentration used (32%).

For NCTC 8325, the addition of sub-inhibitory concentrations of manuka and manuka/kanuka honey significantly enhanced biofilm formation, increasing it by 1.5- and 2-fold, compared to the untreated control ($p < 0.001$). In contrast, Medihoney, clover honey and the sugar solution did not enhance biofilm formation by strain NCTC 8325 at any concentration tested.

*S. aureus* strain ATCC 25923, which is a standard clinical reference strain, produced similar results to NCTC 8325, including the enhancement of biofilm formation following

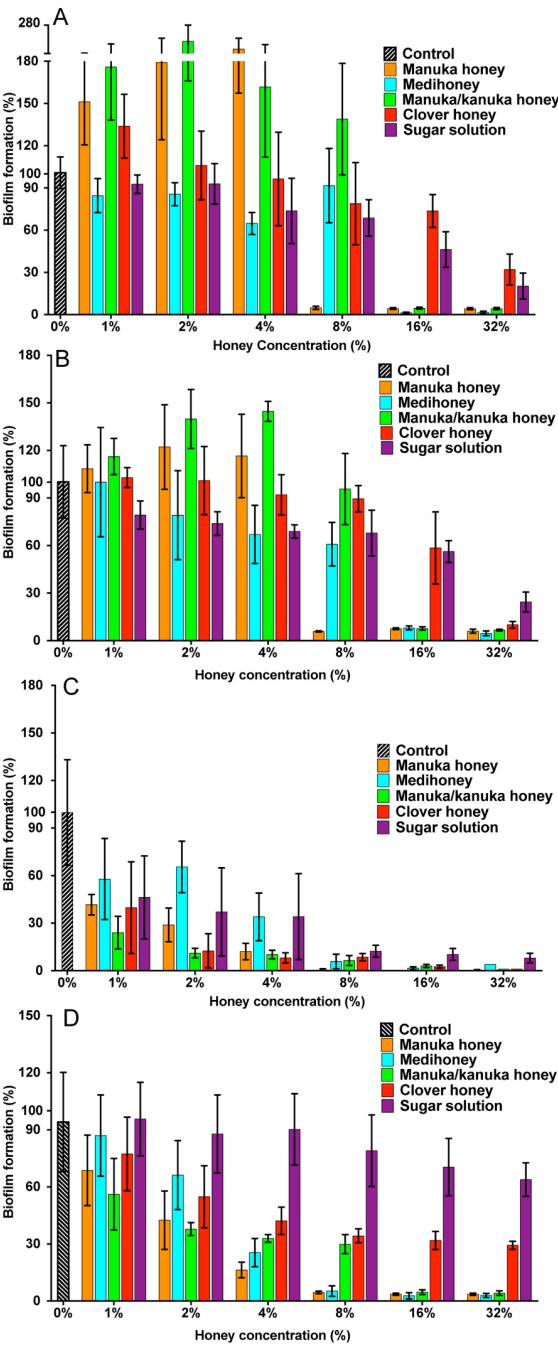

**Figure 3 Effects of NZ honeys and sugar on *S. aureus* biofilm formation.** *S. aureus* biofilms were allowed to form in the presence of four different NZ honey types (manuka, Medihoney, manuka/kanuka or clover) or a sugar solution. Biofilm formation was assessed using a static biofilm formation assay with crystal violet staining to quantify biomass. *S. aureus* strains are: (A) NCTC 8325; (B) ATCC 25923; (C) MW2 (HA-MRSA) and (D) USA300 (CA-MRSA). Biofilm formation is expressed as a percentage relative to that produced by the untreated control, which is set at 100%. Error bars represent ±standard deviation (SD) of three biological samples performed in triplicate. Statistical significance ($p < 0.05$) was assessed by One-Way ANOVA with Tukey test after confirming normality of the data set for each treatment using the D'Agnostino-Pearson normality test.

sub-inhibitory honey treatment (Fig. 3B). The hospital-acquired MRSA strain MW2—the weakest biofilm former out of all four tested strains (Figs. 2A and 2B; $p < 0.05$)—displayed a very sensitive profile to all of the NZ honeys and the sugar solution at all tested concentrations (Fig. 3C). Even with only 1% honey or sugar solution, a ∼50% reduction in biofilm formation was observed for MW2 ($p < 0.001$). At higher concentrations ($\geq 8\%$), all four NZ honeys were significantly more effective than the sugar solution at preventing MW2 biofilms. The other MRSA strain, USA 300, responded similarly to NCTC 8325, with approximately the same concentrations of manuka-type honey being required to reduce biofilm formation by ∼95%. However, unlike NCTC 8325, sub-inhibitory concentrations of manuka-type honey reduced biofilm formation of USA 300 rather than enhancing it (e.g., 4% manuka-type honeys exhibited ∼50–80% biofilm inhibition of USA 300). Moreover, in USA300, biofilm formation was not affected by the sugar solution at any tested concentration.

The results above can be summarize as follows: (i) all three manuka-type honeys are effective at inhibiting biofilm formation of a range of of MSSA and MRSA strains; with (monofloral) manuka honey being generally more effective than the other maunka-type honeys; and (ii) the manuka-type honeys are generally more effective than clover honey and the isotonic sugar solution, although clover honey was just as inhibitory as the manuka-type honeys for the weakest biofilm former, *S. aureus* MW2.

### The effect of MGO on *S. aureus* biofilm prevention

MGO is a principle antibacterial component of manuka honey responsible for its inhibitory effects on the growth of *S. aureus* and other bacterial species. This is evidenced by the correlation between the MGO level and the proportion of manuka-derived honey in a honey blend (*Jervis-Bardy et al., 2011*; *Lu et al., 2013*). To determine whether MGO is solely responsible for the inhibitory effect of the three manuka-type honeys on *S. aureus* biofilm formation, biofilm assays were performed using MGO at equivalent concentrations to those present in each of the manuka-based honey samples, with and without the addition of the sugar solution (Fig. 4). *S. aureus* NCTC 8325 biofilm formation was not significantly ($p > 0.05$) affected by MGO at concentrations equivalent to that present in 1–16% manuka-type honeys. MGO at medium (700 mg/kg) and high (900 mg/kg) levels at the equivalent concentration to 32% manuka-kanuka honey and Medihoney prevented approximately 50% and 75% biofilm formation, respectively ($p < 0.05$). The addition of the sugar solution to MGO at the same levels present in 16% of all three manuka-type honeys, led to a dramatic decrease (∼95%) in biofilm formation.

### The effect of NZ manuka-type honeys on established *S. aureus* biofilms

Bacterial biofilms are usually already established in open, chronic wounds prior to presentation to the clinic for medical treatment. We therefore assessed the ability of the four NZ honeys to remove established biofilms produced by the four strains of *S. aureus*. These results are presented in Fig. 5, with coloured lines showing biofilm mass present following treatment with different concentrations of the various honey types. While

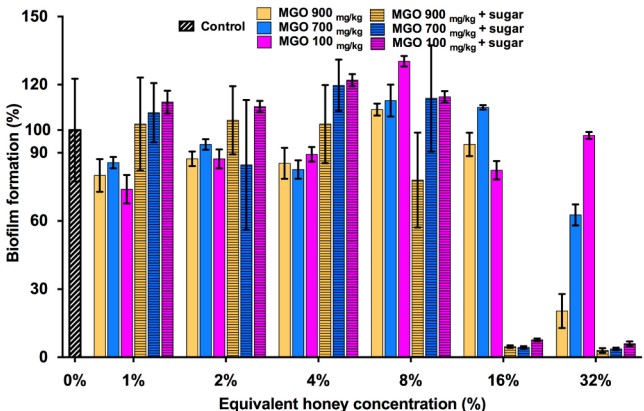

**Figure 4 Effects of MGO on *S. aureus* biofilm formation.** Biofilm formation by *S. aureus* NCTC 8325 grown in the presence of MGO and MGO plus sugar solution. MGO stock solutions were prepared to correspond to the MGO levels in undiluted manuka-type honeys (100 mg/kg of manuka/kanuka honey, 700 mg/kg of Medihoney, and 900 mg/kg of manuka-honey; Table 1). Biofilm formation was assessed using the described static assay with crystal violet staining to quantify biomass. Biofilm formation is expressed as a percentage relative to the untreated control, which is set at 100%. Error bars represent ±standard deviation (SD) of three biological samples performed in triplicate. Statistical significance ($p < 0.05$) was assessed by One-Way ANOVA with Tukey test after confirming normality of the data set for each treatment using the D'Agnostino-Pearson normality test.

there was variation among the *S. aureus* strains in their response to the different honeys, there are some important general trends. First, manuka honey was consistently the most effective at removing biofilm, eliminating almost all of the established *S. aureus* biofilms at concentrations of 16%–32%, ($p < 0.001$ compared to the untreated control sample; Fig. 5 top panel, orange lines). Second, Medihoney and manuka/kanuka honey were also effective at these concentrations for some *S. aureus* strains, but only consistently effective across all four strains at 32% (Fig. 5, blue and green lines). Third, both the clover honey and the sugar solution did not significantly reduce ($p > 0.5$) established biofilm mass until their concentration reached 32%. However, the sugar solution did not remove the USA 300 biofilm, with no significant reduction in biofilm mass at 32% (Fig. 5, purple line).

NCTC 8325, the most efficient biofilm former out of all tested strains, gave a slightly different response toward honey treatment compared to the other three strains. Significant biofilm enhancement occurred in this strain at sub-inhibitory concentrations of manuka honey (1–2%) and manuka/kanuka honey (1–4%) ($p < 0.001$; Fig. 5). In addition, this strain was the least sensitive to the manuka-type honeys. For example, at 8% manuka honey treatment, the NCTC 8325 biofilm mass remained similar to the untreated control ($p > 0.05$; Fig. 5), while the biofilms produced by the other three strains were significantly reduced at this concentration ($p < 0.001$; Fig. 5).

### The effect of NZ manuka-type honeys on cell viability within *S. aureus* biofilms

Elimination of biofilm mass was assessed using crystal violet, a cationic dye that stains all the components of the biofilm. However, this assay cannot assess the viability of cells

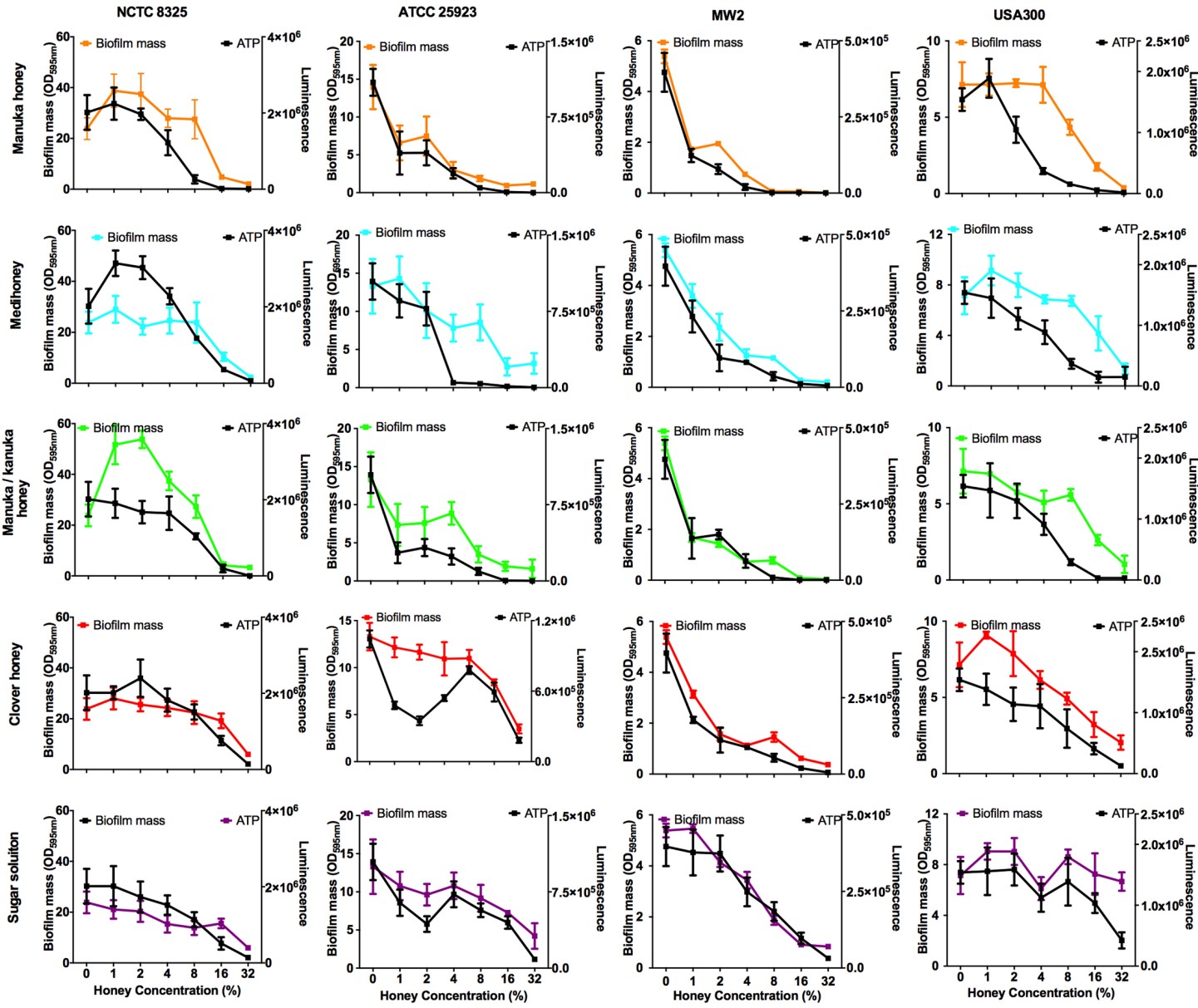

**Figure 5 Effects of NZ honeys on established *S. aureus* biofilms and cell viability within the biofilms.** Established *S. aureus* NCTC 8325, ATCC 25923, MW2 and USA300 biofilms were treated with NZ honeys – manuka, Medihoney, manuka/kanuka, clover, and a sugar solution. The remaining biofilm masses were quantified using crystal violet staining (left *y*-axis) and cell viability within these remaining biofilms were assessed using the BacTitre Glo Viability Kit (right *y*-axis). Error bars represent ±standard deviation (SD) of three biological samples performed in triplicate Statistical significance ($p < 0.05$) was assessed by One-Way ANOVA with Tukey test after confirming normality of the data set for each treatment using the D'Agostino-Pearson normality test.

remaining within the biofilm structure (*Bauer et al., 2013*). To determine this, we used a BacTitre Glo assay, which measures ATP levels as a proxy for viability. Side-by-side CFU measurements showed that the level of ATP detected in these assays was proportional to the count of viable cells per well (ranging from $10^3$ to $10^7$ CFU/well) (Fig. 1).

The viability of cells remaining in the biofilm after the various treatments is shown in Fig. 5. In general, cell viability decreased in proportion to the elimination of biofilm biomass (Fig. 5, black lines). However, several exceptions to this general trend were observed. In some cases, biofilm biomass increased but cell viability did not, e.g., NCTC 8325 biofilms with low concentrations of manuka (2%) and manuka/kanuka honey (1–4%) (Fig. 5). In others, biofilm biomass remained relatively constant while cell viability increased, e.g., NCTC 8325 with 1–4% Medihoney ($p < 0.05$; Fig. 5), and ATCC 25923 with 4% and 8% clover honey. Another deviation from the general trend was a significant reduction of cell viability while biofilm biomass remained unaffected, seen for NCTC 8325 and USA 300 with 4% and 8% manuka honey treatment (Fig. 5; $p < 0.05$). This emphasizes the importance of assessing viability alongside crystal violet assays for biofilm assessment.

Overall, at concentrations easily attainable in the clinic, the tested four NZ honeys were effective at eliminating biofilm biomass and at killing both MSSA and MRSA *S. aureus* cells in the residual biofilm. Among the honey types, manuka honey was the most effective, where the elimination of biofilm biomass largely paralleled the reduction in viability. Following treatment with 8% manuka honey only ~10% of cells were viable in the remaining ATCC 25923 and USA 300 biofilms, compared to the untreated control (i.e., 0% honey), and no generation of ATP could be detected from MW2 (Fig. 5). This is similar to the degree of biofilm biomass removal, where 85–98% of biofilm biomass was removed following 8% manuka honey treatment. Although NCTC 8325 biofilm biomass was seemingly unaffected at 8% manuka honey compared to the untreated control (Fig. 5), the number of viable cells detected within this biofilm was drastically reduced by approximately 80% ($p < 0.001$; Fig. 5).

### The effect of MGO on established *S. aureus* biofilms

To assess the contribution of MGO alone, as well as MGO plus sugar, to biofilm removal, these components were tested on established *S. aureus* NCTC 8325 biofilms (Fig. 6). MGO levels equivalent to the presence of 1–8% manuka/kanuka honey (Table 1) caused biofilm biomass to increase approximately 2-fold, relative to the untreated control ($p < 0.001$). However, the established biofilm biomass was not reduced significantly ($p > 0.05$), for any of the tested concentrations (1–32%) of MGO by itself, or in combination with the sugar solution. Thus, neither MGO nor the combination of MGO with sugar is solely responsible for the elimination of biofilms observed with these manuka-type honeys.

### Visualizing the effects of NZ manuka-type honeys on established *S. aureus* biofilms

To assess the effect of the NZ honeys on *S. aureus* NCTC 8325 biofilms at the cellular level, we used confocal laser scanning microscopy (CLSM) of biofilms stained with fluorescent dyes for the detection of live and dead bacteria. This allows both the visualization of

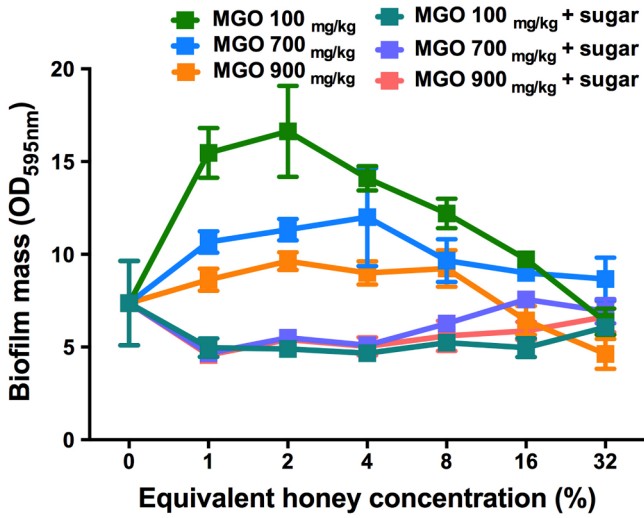

**Figure 6 Effects of MGO on established *S. aureus* biofilms.** *S. aureus* NCTC 8325 biofilms were treated with MGO and a combination of MGO and the sugar solution. MGO stock solutions were prepared to correspond to the MGO levels in undiluted honey (100 mg/kg of manuka/kanuka honey, 700 mg/kg of Medihoney, and 900 mg/kg of manuka honey; Table 1). The crystal violet stained residual biofilm mass after 24 h treatment was quantified using optical density ($OD_{595 nm}$). Error bars represent ±standard deviation (SD) of three biological samples performed in triplicate. Statistical significance ($p < 0.05$) was assessed by One-Way ANOVA with Tukey test after confirming normality of the data set for each treatment using the D'Agnostino-Pearson normality test.

individual cells within the biofilm in three dimensions and the effect of treatments on cell viability to be determined. Treatment by sub-inhibitory (1% and 2%) and inhibitory (16% and 32%) concentrations of NZ honeys was visualized by viewing fluorescently-labelled live (Syto9; green) and dead (propidium iodide; red) cells. Representative images of each treatment are presented in Fig. 7 and quantification of live and dead cell biofilm biomass for several samples for each treatment is shown in Fig. 8. In general, the established biofilm biomass decreased with increasing concentrations of manuka-type honey. More specifically, manuka honeys were effective in reducing the live cells in established *S. aureus* biofilms. Sub-inhibitory concentrations of all the manuka-type honeys (1% and 2%) and the sugar solution did not reduce the amount of biomass compared to the non-treated control cells (Fig. 8). This is shown in Fig. 7 where the untreated control cells displayed a green (live-cell) lawn that covered nearly the entire surface and this remained following treatment with 1% and 2% manuka-type honeys. At concentrations of 16% and 32%, the manuka-type honeys substantially reduced the density and depth of the biofilm, along with the amount of live cells, compared to the untreated control (Figs. 7 and 8). For example, the 32% manuka honey significantly reduced the Syto9 stained (live) biofilm biomass to 10% ($p < 0.001$) compared to the non-treated live biofilm biomass (Fig. 8).

Only small micro-colonies were present following treatment with 32% manuka honey, and the colour of the biofilms was predominantly yellow (where both the green and red dye were retained within cells), indicating mostly dead cells. In contrast, 32% clover honey and sugar solution reduced the total biomass by a maximum of 30% ($p < 0.001$) compared to

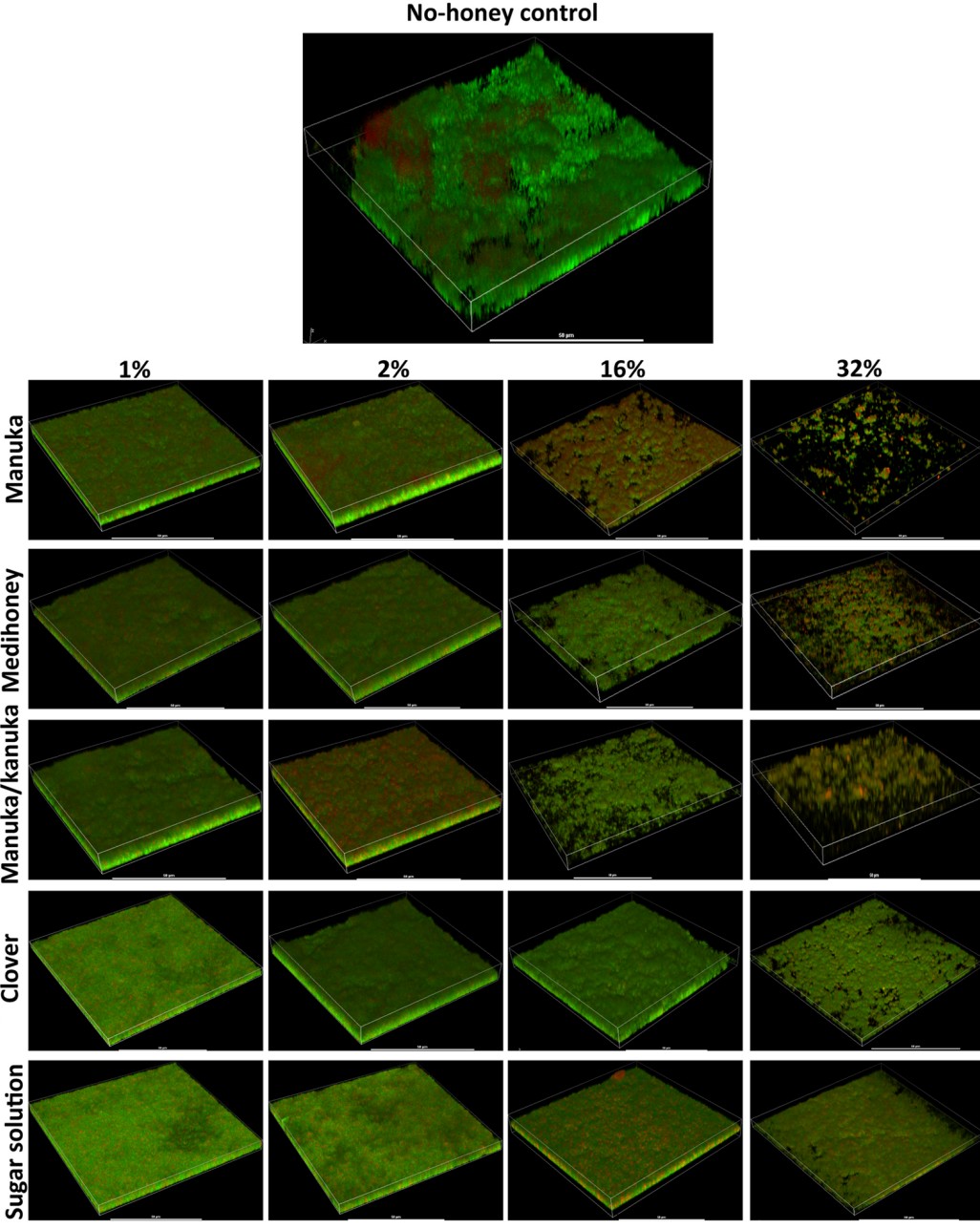

**Figure 7 Live/dead staining of different honey treated established biofilms.** Established biofilms produced by *S. aureus* NCTC 8325 were treated with TSB containing honey (manuka, Medihoney, manuka/kanuka or clover) or sugar solution at 1%, 2%, 16%, and 32% (w/vol). Syto9 (green; viable cells) and propidium iodine (red; dead cells) stained images were acquired using Nikon A1 Confocal Laser Scanning Microscope. The 3D- images were reconstructed using NIS-elements (version 10). Scale bar represents 50 μm.

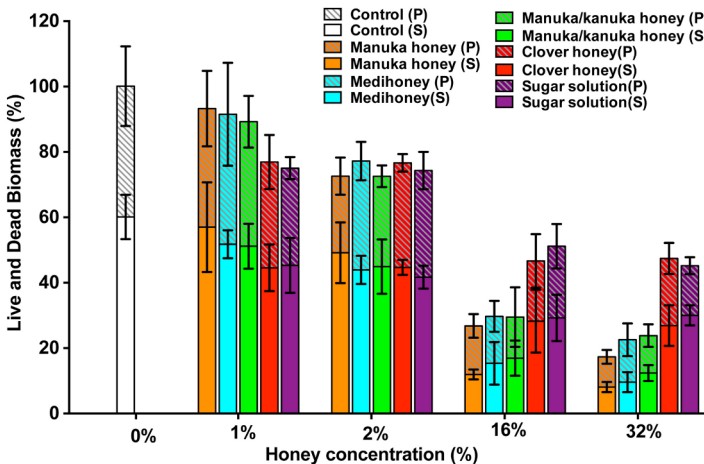

**Figure 8 Quantitative analysis of live/dead stained honey treated biofilms.** The established *S. aureus* NCTC 8325 biofilm was treated with New Zealand honeys (manuka honey, Medihoney, manuka/kanuka honey, and clover honey) and a sugar solution at 1%, 2%, 16%, and 32% (w/v) concentrations. Biofilms were co-stained with Syto9 (S, viable cells) and propidium iodide (P, dead cells) and analyzed using COMSTAT. The estimated live (S) and dead (P) biomass (volume of the biofilm over the surface area ($\mu m^3/\mu m^2$)) are expressed as a percentage of the non-treated control live and dead biomass, which is set at 100%. Error bars represent ±standard deviation (SD) of three biological samples where eight representative images were acquired. Statistical significance ($p < 0.05$) was assessed by One-Way ANOVA with Tukey test after confirming normality of the data set for each treatment using the D'Agnostino-Pearson normality test.

the non-treated control (Fig. 8). This result corresponds to the 3D reconstructed images, where the Syto9 stained cells remained dominant after treatment (Fig. 7). At 32%, clover honey or sugar solution, a substantially larger Syto9 stained (live) lawn remained in comparison to the 32% manuka-type honeys, although the biomass was less confluent than in the untreated control. These results are consistent with the results obtained with the crystal violet stained biofilm biomass and ATP viability assays.

## Assessing the development of resistance to manuka-type honeys in *S. aureus* biofilms

Bacteria that are exposed to sub-inhibitory concentrations of antimicrobial agents generally develop resistance to these agents (*Braoudaki & Hilton, 2004*; *Davies, Spiegelman & Yim, 2006*). The ability of cells released from *S. aureus* NCTC 8325 biofilms to develop resistance after exposure to sub-inhibitory concentrations of honey was investigated, and the results are summarized in Table 3. All cells recovered from the *S. aureus* biofilm after 24 h with 8% of all three manuka-type honeys were viable and able to form biofilms in media (TSB). However, they were unable to grow planktonically when subsequently exposed to 8% manuka, or 16% Medihoney and manuka/kanuka honey (the MIC levels for these honeys). Biofilm formation was also inhibited by 8% manuka honey, and by 16% Medihoney and manuka/kanuka honey. These growth- and biofilm-inhibitory concentrations of manuka-type honeys are the same as those observed for cells that had not previously been treated with these honeys. These results indicating that planktonic cells

**Table 3  Resistance of *S. aureus* cells recovered from biofilms after 8% manuka honey treatments[a].**

| Honey (%) | Type of assay | Manuka honey | Medihoney | Manuka/kanuka honey |
|---|---|---|---|---|
| **0** | Growth | √ | √ | √ |
|  | Biofilm formation | √ | √ | √ |
| **8** | Growth | × | × | √ |
|  | Biofilm formation | × | √ | √ |
| **16** | Growth | × | × | × |
|  | Biofilm formation | × | × | × |

**Notes.**
[a] A tick means that there was normal growth or biofilm formation and a cross means that there was no growth or no biofilm formation.

released from the biofilms with exposure to sub-inhibitory concentrations did not acquire resistance to the same honey treatment.

## DISCUSSION

Chronic wounds are costly and difficult to treat (*Hoyle & Costerton, 1991*; *Ranall et al., 2012*; *Sen et al., 2009*), and bacterial biofilms are important contributors to the delay in healing. Honey is a promising alternative treatment for these wounds, and studies have indicated that it is able to prevent bacterial biofilms and eliminate established biofilms *in vitro* (*Alandejani et al., 2008*; *Maddocks et al., 2013*; *Maddocks et al., 2012*; *Majtan et al., 2013*). However, the effective concentration of honey reported by these studies varies significantly, making it hard to establish a foundation for the efficacy of honey on chronic wound-associated bacterial biofilms in the clinic. This is probably largely due to the fact that, in most of these studies, very little information is reported on the honey itself, including the floral source, geographic location, storage conditions, and the level of the two principle antibacterial components, MGO and hydrogen peroxide. Here we utilize a suite of well-defined NZ honeys, including manuka-type honeys (manuka, Medihoney and manuka/kanuka honey) and clover honey, to investigate their anti-biofilm activity on a range of *S. aureus* biofilms that differ in their ability to form biofilms. We show that manuka-type honeys can be used to kill all MSSA and MRSA cells when present as a biofilm in a chronic wound, supporting the use of this honey as an effective topical treatment for chronic wound infections.

Our study has shown that prevention of *S. aureus* biofilm formation occurred at honey concentrations that also inhibit planktonic growth (Figs. 3A–3D; Table 2), suggesting that biofilm prevention was a consequence of planktonic growth inhibition, as opposed to any specific effects on biofilm development. Other studies have also shown that manuka-type honeys can inhibit bacterial biofilm formation, however, the concentrations required were higher than those reported to inhibit growth (*Alandejani et al., 2008*; *Maddocks et al., 2013*; *Maddocks et al., 2012*; *Majtan et al., 2013*).

We found that higher concentrations of all honeys were necessary to eliminate established biofilms compared to those needed for prevention, as assessed by both quantification of biofilm biomass and cell viability. Manuka honey was the most effective,

closely followed by both Medihoney and manuka/kanuka honey. Elimination of biofilms was visually confirmed using CLSM of fluorescently-stained live and dead cells. The sugar content of honey clearly mediates some effect, as sugar solution and clover honey were able to eliminate established biofilms at high concentrations (32%), as has been shown in other studies (*Chirife et al., 1983*; *Chirife, Scarmato & Herszage, 1982*). However, manuka-type honeys consistently achieved biofilm elimination at lower concentrations, suggesting that components specifically within manuka-type honeys contribute towards biofilm elimination. The concentrations of manuka-type honeys that show significant anti-biofilm activity are easily achievable in the clinic, since honey dressings typically contain >80% honey (*Cooper et al., 2010*).

The use of assays for total biofilm biomass and cell viability to examine the effects of the various treatments on biofilm elimination afforded some other interesting observations. We observed that in some cases, sub-inhibitory concentrations of two of the manuka-type honeys enhanced biofilm formation; however, cell viability did not increase. This could be due to a stress response, as has been previously observed when bacteria are exposed to sub-inhibitory concentrations of antibiotics (*Haddadin et al., 2010*; *Kaplan et al., 2012*; *Mirani & Jamil, 2011*; *Subrt, Mesak & Davies, 2011*). In other cases, no reduction of biofilm biomass was observed but cell viability was significantly reduced. This suggests that unlike antibiotics, the manuka-type honeys (or active components therein) are able to penetrate through the biofilm matrix, killing the bacterial cells whilst leaving intact matrix.

It is believed that MGO is the primary component in manuka-type honeys responsible for its anti-biofilm activity (*Jervis-Bardy et al., 2011*; *Kilty et al., 2011*). The effectiveness of the different manuka-type honeys tested here did increase with MGO content. However, the same degree of biofilm prevention and elimination could not be reproduced with equivalent amounts of MGO either alone or in combination with sugar. In the case of prevention, MGO alone was generally ineffective, although a significant amount of biofilm prevention was achieved in combination with sugar. This suggests that the MGO and sugar do contribute to biofilm prevention, but their effects are not as strong as those observed with manuka honey.

Unlike the three NZ manuka-type honeys, neither MGO alone nor MGO with sugar at honey-equivalent concentrations showed significant *S. aureus* biofilm elimination. This indicates that the ability of manuka-type honeys to eliminate biofilms of this organism is due to one or more components present in the honey other than MGO and sugar, such as low pH, hydrogen peroxide, phenolics and other unknown components (*Jagani, Chelikani & Kim, 2009*; *Jervis-Bardy et al., 2011*; *Kilty et al., 2011*; *Zmantar et al., 2010*). Interestingly, while the kanuka/manuka honey had a relatively high rate of hydrogen peroxide production compared to the manuka and Medihoney (Table 1), but low MGO levels, it was not any more active against biofilms of *S. aureus*. This suggests that, at least for this organism, hydrogen peroxide within these manuka-type honeys does not provide significant anti-biofilm activity.

## CONCLUSIONS

This study is the first to use a suite of well-characterized manuka-type honeys against a range of strains of *S. aureus* that differ in their ability to form biofilms. We demonstrate that: (1) at very low levels, some honeys can enhance biofilm formation, presumably by evoking a stress response similar to that seen with some antibiotics; (2) the ability to prevent or eliminate biofilms is influenced by MGO levels and the presence of sugar, but these alone do not account for all of the anti-biofilm effect; (3) honey is able to reduce biofilm mass and also to kill cells that remain embedded in the biofilm matrix; and (4) planktonic cells released from biofilms following honey treatment do not have elevated resistance to honey. Taken together our results show that if used at an appropriate therapeutic level, manuka-type honey can be used to kill *S. aureus* when present as a biofilm in a chronic wound, supporting the use of this honey as an effective topical treatment for chronic wound infections.

## ACKNOWLEDGEMENTS

CSLM was performed at the UTS Microbial Imaging Facility.

### Funding

Funding sources include: Comvita NZ Ltd, The Australian Research Council (ARC Linkage Project LP0990949), CBW was supported by an Australian National Health and Medical Research Council Senior Research Fellowship (571905). LT was supported by a University of Technology, Sydney Chancellor's Postdoctoral Fellowship. The funders had no role in study design, data collection and analysis, decision to publish, or preparation of the manuscript.

### Grant Disclosures

The following grant information was disclosed by the authors:
Australian Research Council Linkage Project grant: LP0990949.
Australian National Health and Medical Research Council Senior Research Fellowship: 571905.

### Competing Interests

Ralf C. Schlothauer is an employee of Comvita New Zealand (NZ) Limited which trades in medical grade manuka honey (Medihoney). Comvita NZ Ltd. have partially funded the work through a contribution to Linkage Project LP0990949 funded by the Australian Research Council. Chief Investigators on this project include Elizabeth J. Harry, Cynthia B. Whitchurch, Lynne Turnbull, Dee A. Carter and Partner Investigator Ralf C. Schlothauer. Our competing interests do not alter our adherence to all the PeerJ policies on sharing data and materials. We also note that co-author Associate Professor Dee A. Carter is an Academic Editor for PeerJ.

## Author Contributions

- Jing Lu performed the experiments, analyzed the data, wrote the paper, prepared figures and tables, reviewed drafts of the paper.
- Lynne Turnbull and Cynthia B. Whitchurch conceived and designed the experiments, analyzed the data, contributed reagents/materials/analysis tools, reviewed drafts of the paper, revised the manuscript.
- Catherine M. Burke analyzed the data, wrote the paper, reviewed drafts of the paper.
- Michael Liu analyzed the data, wrote the paper, reviewed drafts of the paper.
- Dee A. Carter analyzed the data, contributed reagents/materials/analysis tools, wrote the paper, reviewed drafts of the paper.
- Ralf C. Schlothauer conceived and designed the experiments, analyzed the data, contributed reagents/materials/analysis tools, revised the manuscript.
- Elizabeth J. Harry conceived and designed the experiments, analyzed the data, contributed reagents/materials/analysis tools, wrote the paper, reviewed drafts of the paper, revised the manuscript and coordinated the research.

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
