# Peer review of "Manuka-type honeys can eradicate biofilms produced by Staphylococcus aureus strains with different biofilm-forming abilities"

_PeerJ, doi:10.7717/peerj.326_

## Round 0.1 · original submission · Minor Revisions

I agree that the provision of more quantitative data to describe the biofilm structures (taken from more than just a 'representative' image) would make the paper stronger and benefit the readers of your manuscript.

Reviewer 1 ·

Basic reporting

The authors compare the activity of honey samples against
bacterial biofilms and to examine
their effect on viability and biofilm mass.
According to their results, Manuka-type is the most promising honey
to treat biofilm-related infections of S. aureus. This is a detailed study that is interesting for readers.

Experimental design

line 133: The authors should explain why the strain NCTC 8325 produce optimal biofilm without glucose.

line 187: The authors should show the standard curve to demonstrate the correlation between bacterial cell numbers and the luminescence.

line 251 : biofilm- forming ability are assessed after 24h or 48h , why not at 6h or 12h? Could you include some literature about this.

Validity of the findings

No Comments

Additional comments

No Comments

Reviewer 2 ·

Basic reporting

The submission adhere to all PeerJ policies.
Minor issues:
1replace MGO with methylglyoxal in the abstract
2 replace elimination with eradication, line 169 pag. 7

Experimental design

No comments
Minor issue: clear the diluent used to dissolve all samples, line 108 page 5

Validity of the findings

No comments

Additional comments

This paper presents some interesting findings about manuka honeys in the developing of novel anti staphylococcal biofilm treatment.
Because the high complexiity of chemical compostion of honey samples, I believe there are interactive and synergistic effects among the different components (the two principal and other) that result in the observed biological activity.

·

Basic reporting

The article is well written and introduces all relevant information. The methods are, in the main, easy to follow (but see points 3 and 4). I found the tables, figures and legends to be clear and understandable, despite the complexity of Figures 3 and 5, which contain numerous items. The figures and tables are relevant (but see point 2). The conclusions are supported by the evidence presented, but see point 5 regarding the presentation of more quantitative data.


1. The first results section is titled
“The Effect of NZ manuka-Type Honeys on S. aureus Biofilm Formation.”
The first part deals with planktonic cells and is sufficient for a section entitled
“The Effect of NZ manuka-Type Honeys on the planktonic growth of S. aureus.”

2. Does an online journal like PeerJ really need supplementary material? If it is relevant put it in the paper, if it isn't don't. All the supplementaries do is create extra files for the reader to (forget to) download.

Experimental design

3. Crystal Violet (CV) vs ATP assays. CV stains all biomass (live and dead cells and the biofilm matrix); ATP assays are a measure of the viability of the biofilm cells. The facts of this come out eventually but it would be nice to see this also stated at the beginning of the relevant methodology sections, perhaps with justification for using each assay. This would be at the start of Biofilm formation assays (p6) and Determination of Bacterial Cell Viability within Biofilms (p8).

4. Regarding the ATP assay method (p8). The ATP is in the cells, these are in the biofilm attached to the microplate surface; I think that a note in the method stating that the kit reagents lyse/permeablise these cells to release ATP as a substrate for the luciferase also present in the kit reagent would be helpful to those readers not familiar with the assay- (hopefully that is how it works).It would be helpful to include data for the standard curve in the methods on p8, I think this is the figure in S2 eventually referenced on line 341, p15.

Validity of the findings

5. Imaging. The reconstructed confocal images represent the effect of honey on S. aureus biofilms well, but there are only single images for each treatment. A more convincing representation would also tabulate parameters e.g. thickness, surface coverage, biofilm volume (volume per unit area, perhaps for live cells and dead cells). The authors will have the data for this analysis, although the 'representative' nature of the images is not quantified. I would see this as a minor addition to the paper in terms of additional work required, but an addition that would substantially strengthen the argument presented.

6. The images in Figure 5 show biofilm stained live vs dead. The 'representative' images suggest to me that at lower honey concentrations there is cell death in the biofilm, and at higher concentrations these dead cells have gone and the biofilm has reduced surface coverage. Do the authors agree? Perhaps this is worthy of comment. The model used is flow through, which may help remove dead areas of biofilm. Wounds will have static biofilms without flow, and so the dead cells may remain in the biofilm, possibly participating as part of a matrix honey antimicrobials would have to penetrate.

7. A final point that does not come out in these images is: how well does the honey antimicrobial penetrate to kill cells?, and, if available, data localising the dead zone may be helpful. The question really is whether antimicrobial penetration is due to sequential killing of surface layers, which are then removed by the flow, or due to full penetration to all biofilm areas. Microplate assays suggest there is full penetration of static biofilms, assuming the static microplate biofilms are of equivalent thickness when compared to the flow cell biofilms. So, a data set has been obtained using live/dead staining that could possibly be analysed further to provide additional insight. If this is not possible, or is perhaps a difficult analysis for meaningful data, a note in the discussion would be valuable. A consideration of the limitations and advantages of the biofilm conditions used, especially flow vs static, in modelling biofilm infections would also be a valuable topic for discussion.

---

## Round 0.2 · Minor Revisions

Apologies for not spotting this before, but it is not appropriate to use the standard error (SEM) as a measure of variation. The SEM is an estimate of how far the sample mean is likely to be from the population mean. Instead you should use the standard deviation which is the degree to which individuals within the sample differ from the sample mean.

Furthermore, can you please ensure that all Figure legends clearly explain what statistical analysis was performed, rather than just stating the p values. Were samples tested for normality first?

---

## Round 0.3 · Minor Revisions

Apologies, there are still a couple of minor issues that need to be resolved before acceptance. In the key for Figure 3, there appears to be a spelling mistake and manuka appears as manuk. Finally there is no key indicating what the colour bars are in Figure 8. The way the columns are arranged in this figure is also very confusing as there are lots of gaps so it is unclear which bars belong to which honey concentration.

---

## Round 0.4 · accepted · Accept

Thank you for making all the corrections requested.